# Tall Cell Carcinoma with Reversed Polarity: Case Report of a Rare Special Type of Breast Cancer and Review of the Literature

**DOI:** 10.3390/biomedicines11092376

**Published:** 2023-08-24

**Authors:** Maiar Elghobashy, Stephanie Jenkins, Zachary Shulman, Anne O’Neil, Sofia Kouneli, Abeer M. Shaaban

**Affiliations:** 1The Royal Wolverhampton NHS Trust, Wolverhampton WV10 0QP, UK; 2University Hospitals Plymouth NHS Trust, Plymouth PL6 8DH, UK; 3Queen Elizabeth Hospital Birmingham, University of Birmingham, Birmingham B15 2TT, UK

**Keywords:** breast cancer, reversed polarity, tall cell carcinoma

## Abstract

**Background:** Tall cell carcinoma of the breast with reversed polarity (TCCRP) is a rare type of invasive breast cancer with overlapping features with papillary thyroid carcinoma and a characteristic molecular profile. Few cases have been reported in the literature since the first case was described in 2003. **Case presentation:** We present the case of a 41-year-old female with a symptomatic left breast lump. Image-guided core biopsy was diagnosed as triple-negative apocrine carcinoma. Surgical excision revealed an invasive carcinoma with solid papillary pattern, nuclei arranged away from the basement membrane (reversed polarity) and luminal eosinophilic colloid-like material. The tumour was GATA3-, CK5-, CK14- and CK7-positive and TTF1-negative. Specialist opinion and the identification of hotspot mutations in the *IDH2* p.Arg172 gene via PCR confirmed the diagnosis of TCCRP. **Conclusions:** TCCRP is a relatively recently recognised papillary epithelial neoplasm with characteristic morphological features and molecular profile. Due to its rarity, TCCRP can be diagnostically challenging, and features can be mistaken for benign and malignant lesions. Accurate diagnosis is important in effective treatment of this indolent malignant triple-negative breast cancer, which carries an excellent prognosis.

## 1. Introduction

Tall cell carcinoma of the breast with reversed polarity (TCCRP) is a rare type of invasive breast cancer that has recently been described in the WHO Blue Book as a special type of invasive breast cancer with a characteristic molecular profile [1]. This tumour was also known as ‘breast tumour resembling tall cell variant of papillary thyroid carcinoma’ due to their similar histological appearances. Other previous, not currently recommended, terminologies include solid papillary breast carcinoma resembling the tall cell variant of papillary neoplasms and solid papillary carcinoma with reverse polarity [2,3]. While it morphologically overlaps with papillary thyroid carcinoma, this primary breast carcinoma does not express thyroid-specific markers. It also mimics a number of benign and malignant breast lesions, including intraduct papilloma with epithelial hyperplasia, solid papillary carcinoma and encapsulated papillary carcinoma, and, therefore, this entity can be challenging for the practising pathologist. Due to its rarity, epidemiological data on its prevalence are sparse, with a limited number of cases overlapping in different publications. In addition, the limited number of cases published overlapped in different publications.

The hypothesised pathophysiology of this tumour relates to hotspot mutations in the *IDH2* p.Arg172 gene, which was identified in 84% of reported cases and is not found in other types of invasive breast carcinomas [4]. These mutations are somatic mutations and change amino acid complexes in the isocitrate dehydrogenase 2 enzyme. This enzyme is essential in the citric acid/Krebs cycle, which is a series of reactions that release stored energy. Mutations in this gene have also been found in acute myeloid leukaemia, cholangiocarcinoma and chondrosarcoma [1].

While hormone receptors and HER2 are commonly negative in TCCRP, the tumour has an indolent course, and it is, therefore, important to distinguish it as a subtype of triple-negative breast cancer (TNBC) with favourable prognosis [5].

In this report, we present a rare case of TCCRP of the breast in a 41-year-old patient. The histological differentials are considered, and an up-to-date literature review is presented.

## 2. Clinical Presentation

A 41-year-old female presented symptomatically with a lump in her left breast. There was no previous history other than a treated right breast abscess. Her bilateral mammograms three years earlier were unremarkable. She did not complain of any hormonal symptoms (no changes to the menstrual cycle) and did not have any unintentional weight loss. She had no family history of breast cancer; however, her sister has a rare myeloproliferative disease. On clinical examination, a hard irregular lump in the lower outer quadrant of the left breast measuring 2–3 cm in size was palpated. A mammogram showed that the right breast was normal; however, an asymmetric density in the left lower outer quadrant was noted (BIRADS M4) (Figure 1a). Digital breast tomosynthesis confirmed an ill-defined mass measuring approximately 28 mm. An ultrasound scan confirmed the presence of a suspicious ill-defined mass (U4) with normal axillary nodes (Figure 1b).

A left breast image-guided core biopsy revealed a grade 2 invasive carcinoma, which was interpreted as apocrine carcinoma (B5b). The oestrogen receptor (ER) was negative, and the human epidermal growth factor receptor 2 (HER2) was negative, but the result was not available at the initial multidisciplinary team (MDT) meeting discussion. The patient was given the option of receiving neoadjuvant chemotherapy or chemotherapy post-operatively. As per her request, the patient underwent a wide local excision and sentinel lymph node biopsy, a decision supported by the MDT team.

Macroscopic examination revealed a well-defined pale tumour measuring 22 mm. A microscopically well-demarcated neoplasm showing foci of papillary architecture was confirmed. It comprised a proliferation of large, tall cells that exhibited solid nests with central fibrovascular cores and associated luminal histiocytes. The cells had eosinophilic cytoplasm and vesicular nuclei with inconspicuous nucleoli (Figure 2a,b). Cystic spaces containing luminal eosinophilic secretions resembling thyroid colloid were noted (Figure 2c). One striking feature was the reversed polarity of the nuclei of the tall columnar cells with their nuclei located at the luminal surface, rather than the basement membrane in several areas (Figure 2d–f). Focal luminal calcifications were also noted. No lymphovascular invasion was present, and sentinel lymph nodes were negative. The morphological appearances suggested the very rare diagnosis of TCCRP, and the case was, therefore, sent for expert opinion.

The carcinoma was strongly positive for CK7 (Figure 3c) and negative for ER and HER2. No myoepithelial layer was identified around the lesional structures through p63 (Figure 3b) and CK5 staining. GATA3 immunohistochemistry showed moderate nuclear positivity, but the staining intensity was less than that of the enclosed normal mammary epithelium (Figure 3d). CK5 immunohistochemistry showed patchy positivity, indicative of a basal phenotype (Figure 3a). Calretinin was also positive (Figure 3e). TTF1 was negative. Molecular testing was initiated. *IDH* testing, performed using the QIAGEN therascreen *IDH1/2* RGQ real-time PCR kit, confirmed the presence of a mutation at codon 172 of the *IDH2* gene, which is specific for TCCRP, thus confirming the diagnosis.

The patient received adjuvant radiotherapy with a boost to the tumour bed as per oncology recommendations. No adjuvant chemotherapy was given. The follow-up protocol is for annual surveillance mammograms for five years. However, as the patient is 41 years of age, annual surveillance mammograms will continue until she becomes eligible for the NHS breast-screening programme at age 50. Follow-up has been uneventful, and the patient remained well 19 months following surgery.

## 3. Discussion

This is an extremely rare special type of invasive breast cancer, with a wide age range, which can be misinterpreted, particularly on small core biopsy samples, as metastatic papillary thyroid cancer or as other types of primary cancer, such as apocrine carcinoma. Since its first report in the literature by Eusebi et al. [6], it is thought that up to 80 cases of TCCRP have been reported in the literature (Table 1) [7], all of which occurred in women. The diagnosis was confirmed by identifying the specific *IDH2* gene mutation via PCR testing of a representative tumour paraffin block. A summary of TCCRP cases published in the past 10 years with their immunoprofile is presented, in comparison to the current case, in Table 1.

The above tumours share common histological features; they comprise demarcated columnar epithelial nests surrounded by dense fibrous stroma and often containing fibrovascular cores, resulting in a papillary appearance. Reversed polarity is also noted, where the nuclei are at the apical rather than the basal poles of the epithelial cells [1]. This feature can be patchy or prominent and should be looked for, particularly in bland-looking triple-negative breast cancers. The tumour ducts often contain abundant eosinophilic cytoplasm. These lesions are often triple-negative with a low proliferative Ki-67 index and are frequently CK5/6- and CK7-positive. The use of basal cytokeratins (such as CK5/6 and CK14) is, therefore, useful to confirm a basal phenotype and to differentiate from other mimics. The absence of a surrounding myoepithelial layer via smooth muscle immunohistochemistry (such as smooth muscle myosin; SMM and p63) confirmed the invasive nature.

Since its original description, TCRRP has been noticeable for its resemblance to papillary carcinomas of the thyroid. The latest WHO classification of Endocrine and Neuroendocrine Tumours recognises the tall cell variant of papillary thyroid carcinoma (TC-PTC) as showing a papillary architecture and characteristic nuclear features [15]. The nuclear features of papillary thyroid carcinoma include oval, frequently overlapping, optically clear, nuclei with nuclear pseudoinclusions and intranuclear grooves. The nuclear pseudoinclusions and grooves have also been described in breast TCCRP. Round concentric calcifications (psammoma bodies) are usually seen within the papillae, stroma or lymphocytic infiltrate in up to 50% of papillary thyroid neoplasms. Squamous differentiation can be seen in TC-PTC and may be extensive, a feature not described in TCCRP. Immunohistochemistry should also differentiate PTC from TCCRP. In the confirmation of this diagnosis, it is essential to use immunohistochemical stains for thyroid-specific antigens (such as TTF1 and thyroglobulin) to exclude TC-PTC. The breast origin is supported by the often-strong positive expression for GCDFP-15 and GATA-3. GATA3 has been shown to be a sensitive marker of breast cancer. It is negative in the normal thyroid, and its expression was reported to be extremely low in thyroid cancers (4/126 cases; 3.2%), with no expression noted in the papillary thyroid carcinoma [16]. TCCRP usually also displays strong positivity with anti-mitochondrial antibody. Papillary thyroid carcinomas are positive for TTF-1, thyroglobulin and PAX8. Basal cytokeratins, such as CK5 and CK14, are not expressed in PTC, while broad-spectrum cytokeratins are positive in both tumours. The molecular alterations are also different between both lesions, with driver mutations in the MAPK pathway representing the commonest mutations in PTC. BRAF p.V600E is common in classic PTC and its variants. The diffuse cytoplasmic expression of BRAF p.V600E through immunohistochemistry using the VE1 clone is highly sensitive and specific for the detection of the mutation in PTC. Molecular studies to identify the *IDH2* gene mutation can exclude a thyroid origin for these tumours [1]. The differentiation is important as TCCRP lesions carry a favourable prognosis due to their indolent clinical course, with only occasional cases reported to be associated with nodal/distant metastasis, despite its triple-negative phenotype [1,17]. Immunohistochemistry for *IDH2* has also been reported as a highly sensitive marker for diagnosis, detecting 93% of the molecularly confirmed cases. It was not performed in the current case, and molecular testing was regarded as a more specific test for confirming the specific mutation.

The current tumour was diagnosed as apocrine carcinoma on core biopsy. TCCRP mimics several benign and malignant breast lesions, including apocrine carcinoma, secretory carcinoma, solid papillary carcinoma, encapsulated papillary carcinoma, infiltrating epitheliosis and florid epithelial hyperplasia of the usual type. Table 2 summarises the diagnostic criteria, immunohistochemical and molecular profile of the common histological mimics. Infiltrating epitheliosis is a rare complex sclerosing benign lesion with fibroelastotic stroma and florid epithelial proliferation of bland cells, reminiscent of epithelial hyperplasia of the usual type [18]. The infiltrative nature of the lesion and the lack of surrounding myoepithelium in places may result in a mistaken diagnosis of malignancy. Infiltrating epitheliosis and florid epithelial hyperplasia can be distinguished by the bland morphology, mixed luminal and basal cytokeratin profile and preservation, at least focally, of a surrounding myoepithelial layer [18,19]. While infiltrating epitheliosis may show attenuation of the myoepithelial layer in places, it is usually preserved in areas. Due to the rarity of TCCRP, the practising pathologist may not be aware of the diagnostic features, leading to a misdiagnosis. Referral for specialist opinion is good practice in doubtful cases. If the diagnosis is suspected morphologically, molecular testing for p.Arg 172 *IDH2* gene mutation can be performed on a representative paraffin block to confirm the diagnosis.

## 4. Conclusions

We present the 80th case of TCCRP. The hallmark is a papillary pattern with reversed polarity of the nuclei, a morphological feature that can be patchy or focal. The diagnosis should be confirmed by identifying the p.Arg 172 mutation of the *IDH2* gene. Few cases have been reported since it was first recognised in the literature in 2003. Accurate diagnosis of TCCRP is essential in the treatment of the condition, which carries a much more favourable prognosis compared to standard triple-negative breast cancer, and both neoadjuvant and adjuvant chemotherapy can safely be omitted.

## Figures and Tables

**Figure 1 biomedicines-11-02376-f001:**
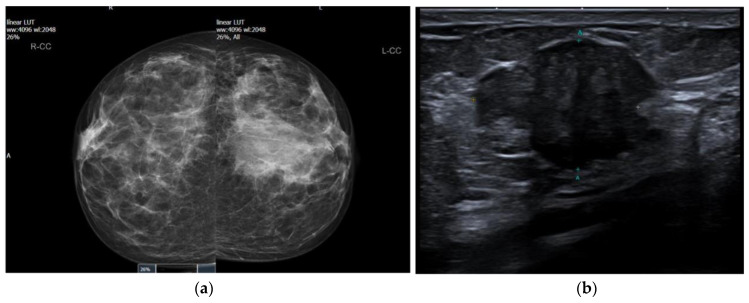
Imaging features of tall cell carcinoma with reversed polarity (TCCRP). (**a**) Asymmetric density in the left breast on mammography. (**b**) Ill-defined mass of the left breast on ultrasonography.

**Figure 2 biomedicines-11-02376-f002:**
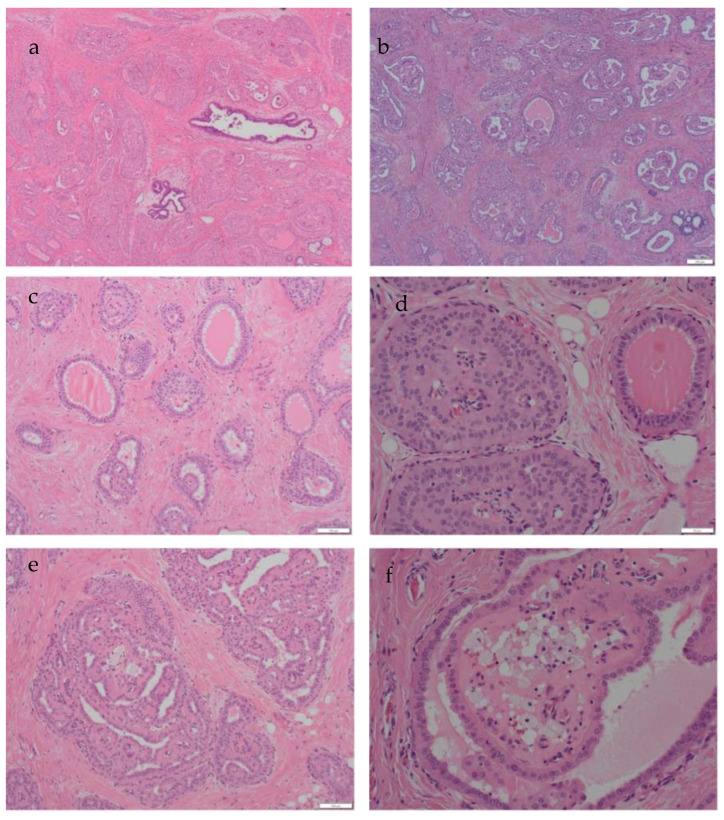
Histological appearances of tall cell carcinoma with reversed polarity. (**a**,**b**) Overall views of the tumour showing pale and eosinophilic cells arranged in solid papillary architecture. There is associated luminal, eosinophilic, colloid-like material (magnification ×40). (**c**) The eosinophilic, colloid-like, luminal secretions are a prominent feature (magnification ×40). (**d**) Tumour cell nuclei are arranged away from the basement membranes and towards the glandular lumina (reversed polarity). There is associated luminal secretion (magnification ×100). (**e**,**f**) The papillary architecture, luminal secretions and reversed polarity of the nuclei are evident (magnification ×100 and ×200).

**Figure 3 biomedicines-11-02376-f003:**
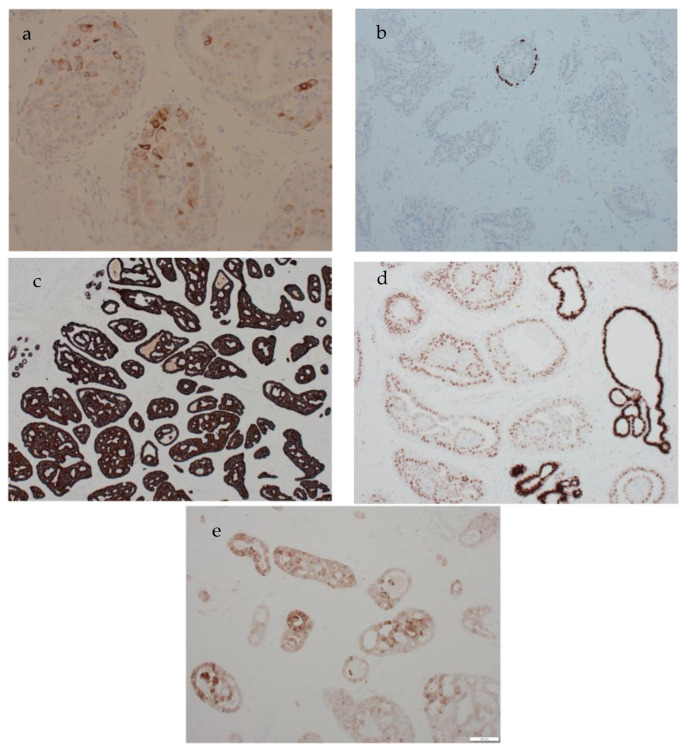
The immunohistochemical profile of tall cell carcinoma with reversed polarity. (**a**) The lesion shows patchy cytokeratin 5 positivity in keeping with a basal phenotype. No myoepithelial layer is seen around the papillary structures confirming an invasive carcinoma (magnification ×200). (**b**) P63 immunohistochemistry confirms the absence of myoepithelium around lesional papillary structures (magnification ×40). The carcinoma is strongly CK7-positive (magnification ×100) (**c**), GATA3-positive (magnification ×100) (**d**) and calretinin-positive (magnification ×100) (**e**).

**Table 1 biomedicines-11-02376-t001:** Summary of case reports/case series of tall cell carcinoma of the breast with reversed polarity in the past 10 years.

Author	Year	Age of Patient	Presenting Features	Radiological Features	Pathology Features	Immunohistochemistry
Elghobashy et al. (current study)	2023	41	Symptomatic lump	Ill-defined mass with normal axillary nodes	Proliferation of large, tall cells that exhibited solid nests with central fibrovascular cores and associated luminal histiocytes	+ve: CK5, CK7, GATA-3, calretinin −ve: ER, PR, HER2, TTF1, p63
Trihia et al., 2021 [8]	2021	70	Incidental mammographic finding	Well-defined 9 mm lesion with normal axillary nodes	Nests of epithelial cells with focal solid papillary pattern. Tall columnar cells with reversed polarity	+ve: GATA3, Keratin 903, mammoglobin (focal) −ve: P63, SMA, CK5/6, ER/PR
Matute et al. [9]	2021	63	Identified via screening	Focal asymmetry, nodularity	Trabeculae, nests of columnar cells with histiocytes, wide granular cytoplasm	+ve: CK5/CK6, CK7, GCDFP-15, GATA-3 −ve: HER2, ER, PR, AR
Jassim et al. [10]	2021	40	Lump	BIRADS IV lesion	Nests of neoplastic cells, solid papillary pattern lined with columnar epithelial cells	+ve: CK5/CK6, GATA3, ER (weakly) −ve: HER2, BCL2
Wei et al., 2021 [11]	2021	72	Palpable mass	Well circumscribed hypoechoic lesion	Papillary architecture, eosinophilic cytoplasm	+ve: ER, GATA-3 −ve: SMMHC, p63, PR
		70	Palpable mass	Nodular density	Reversed polarity, eosinophilic cytoplasm, ovoid nuclei	+ve: CK5, calretinin, ER (weakly), SOX-10, GCDFP-15 −ve: SMMHC, p63,
Zhang et al. [7]	2021	45	Breast mass	Hypoechoic lesion	Circumscribed tumour cell nests with fibrovascular cores	+ve: CK5/CK6, GATA3, GCDFP-15, mammaglobin −ve: ER, PR, HER-2, S-100, p53
Haefliger et al. [12]	2020	60	Breast nodule	BIRADS 4 lesion	Epithelial lesion with nests and solid papillary architecture	+ve: CK5/CK6, GATA3, E-cadherin −ve: ER, PR, HER-2, S-100, TTF1
Foschini et al. [2]	2017	13 cases ranging from 48 years to 85 years	3 cases identified via screening 10 cases with palpable nodules	-	Multilobular architecture with neoplastic cells arranged in papillary, solid and follicular structures, fibrovascular cores	+ve: CK7 (in 10 out of 13 cases), CK14 (in 3 out of 13 cases), CK5/CK6 (in 7 out of 13 cases), GATA-3 (in 5 out of 13 cases) −ve: ER (in 10 out of 13 cases), PR (in 10 out of 13 cases), HER-2 (in all 13 cases)
Bhargava et al. [13]	2017	65	Breast mass	-	Nodular papillary lesion	+ve: CK5/CK6, S-100 −ve: SMMHC, p63, ER
		77	Breast mass	-	Lobulated papillary lesion intersected by thick fibrous bands, eosinophilic secretions	+ve: CK5/CK6, ER (weakly), AE1/AE3, S-100, GATA-3 −ve: p63, SMMHC, TTF-1, thyroglobulin
		48	Breast mass	-	Proliferative nodular lesion, nuclear grooves, reversed polarity	+ve: CK5/CK6, S-100 −ve: p63, SMMHC
Chiang et al. [3]	2016	13 cases ranging from 51 years to 79 years	-	-	Circumscribed nodules of columnar epithelial cells with many containing fibrovascular cores and reverse polarity	+ve: CK5/CK6 (in 12 out of 13 cases), CK7(in 11 out of 13 cases) −ve: p63 (in all cases), SMMHC (in 12 out of 13 cases), ER (in 8 out of 13 cases), PR (in 11 out of 13 cases), HER-2 (in 10 out of 13 cases), TTF-1 (in 12 out of 13 cases)
Colella et al. [14]	2014	79	Breast mass, bloody nipple secretion	Lesion suspicious of malignancy	Cystic spaces containing eosinophilic material, epithelial cells with columnar configuration	+ve: GCDFP-15 −ve: TTF-1, thyroglobulin

**Table 2 biomedicines-11-02376-t002:** Histological features and molecular profile of TCCRP and its malignant mimics.

Feature	Apocrine Carcinoma	Secretory Carcinoma	Solid Papillary Carcinoma	Encapsulated Papillary Carcinoma	Papillary Thyroid Carcinoma	TCCRP
Papillary architecture	Uncommon, may be focal	Uncommon	Common, solid papillary enclosing sinusoidal vessels	Common well developed papillary architecture with cystic change and haemorrhage	Common	Common
Secretion	Not a feature	Common including intra and extracellular PAS positive secretion	Extracellular mucin is common	Not a feature	Common, follicles with colloid	Common, colloid- like
Cells	Large with vesicular nuclei and ample granular eosinophilic cytoplasm. Can be low, intermediate of high grade	Usually low to intermediate grade nuclei, Inconspicuous nucleoli, prominent secretion	Low to intermediate grade nuclei, inconspicuous nucleoli	Low to intermediate grade nuclei, infrequent mitoses	Oval crowded pale nuclei with nuclear grooves and pseudo includsions	Pale cells, low to intermediate grade nuclei, inconspicuous nucleoli
Nuclear polarity	Normal	Normal	Normal	Normal	Normal	Reversed
Immunohistochemistry	GCDFP-15, GATA-3, AR positive. Often ER/PR negative, can be HER2 positive	Often triple negative, NTRK positive, alpha1 antitrypsin positive	Positive for neuroendocrine markers, usually lacking myoepithelial staining in and around lesion	Usually lacking myoepithelial staining in and around lesion. Negative for neuroendocrine markers	Positive for TTF1, Thyroglobulin, PAX8, CK7, AE1/3, CAM5.2	Positive for low and high molecular weight cytokeratins, Often triple negative, GATA-3 positive, antimitochondrial antibody positive *IDH2* positive
Molecular profile	Non-specific	ETV6-NTRK3 gene fusion	PIK3CA in 45% of cases	Nonspecific	*BRAF, RET* mutations	*IDH2* p.Arg172 mutations

## Data Availability

No new data were created or analyzed in this study. Data sharing is not applicable to this article.

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
