# Peer review of "Tall Cell Carcinoma with Reversed Polarity: Case Report of a Rare Special Type of Breast Cancer and Review of the Literature"

_biomedicines, 2023, doi:10.3390/biomedicines11092376_

Round 1

Reviewer 1 Report

Very well presented case report including a very useful literature review on the breast tumour that is now known as Tall cell carcinoma with reversed polarity. Although this a rare tumour, pathologist awareness of this entity is important to avoid misinterpretation as a more aggressive triple negative breast carcinoma or a benign lesion. The report includes a detailed description of the key pathological features, immunohistochemistry and molecular characteristics. 

These tumours also typically display positivity with anti-mitochondrion antibody.  This is not essential for diagnosis but it would be useful to include in the section on immunohistochemistry. 

In general the paper is very well written.

There are a few very minor edits that would enhance the text:

Line 37 - data  ..... are (not is)

Line 68 - delete but before the HER2 comment

Line 74 - verb missing - suggest inserting revealed or displayed  

Line 121 - insert that before up

Lines 134 and 135 - multiple use of the word often - suggest replacing one with frequent 

Line 136 - perhaps essential rather than quintessential 

Table 1 - calretinin is spelt as calretinen in first row.

Author Response

Response to reviewer 1 comments

We thank the reviewers for their valuable comments that improved the content and readability of the manuscript. All edits have been addressed point by point as below. All changes in the manuscript have been made in red for easy identification.

Reviewer 1 comments

Comments and Suggestions for Authors

  • Very well presented case report including a very useful literature review on the breast tumour that is now known as Tall cell carcinoma with reversed polarity. Although this a rare tumour, pathologist awareness of this entity is important to avoid misinterpretation as a more aggressive triple negative breast carcinoma or a benign lesion. The report includes a detailed description of the key pathological features, immunohistochemistry and molecular characteristics. 

We thank the reviewer for the positive comments.

  • These tumours also typically display positivity with anti-mitochondrion antibody.  This is not essential for diagnosis but it would be useful to include in the section on immunohistochemistry. 

That is correct and this information has been added to the immunohistochemistry section and also in the characteristic immunoprofile of TCCRP in Table 2.

Comments on the Quality of English Language

  • In general the paper is very well written.
  • There are a few very minor edits that would enhance the text:
  • Line 37 - data  ..... are(not is)
  • Line 68 - delete butbefore the HER2 comment
  • Line 74 - verb missing - suggest inserting revealed or displayed
  • Line 121 - insertthat before up
  • Lines 134 and 135 - multiple use of the word often - suggest replacing one with frequent
  • Line 136 - perhaps essentialrather than quintessential 
  • Table 1 - calretinin is spelt as calretinen in first row.

All the above edits have been made. All are in red to spot easily.

We hope we have addressed the reviewers’ comments satisfactorily and look forward to your decision.

Reviewer 2 Report

The manuscript by Elghobashy and co-workers focuses on the tall cell carcinoma of the breast with reversed polarity (TCCRP). The Authors describe a new case of this rare cancer and summarize the current knowledge about TCCRP. The article is interesting and well-written. I would suggest some changes as listed below.

[1] Please discuss the similarities/differences between TCCRP and papillary thyroid cancer in the context of the latest WHO histologic classification of thyroid tumors released in 2022.

[2] I wonder if the lump in this patient had a specific value on the BIRADS scale. If so, please provide this information in the manuscript.

[3] Gene names should be italicized (e.g., p. 4, line 123).

[4] It seems to me that the Reader can expect a better explanation of the meaning/purpose of testing the various antigens mentioned in the manuscript in the context of breast/thyroid cancers (e.g. GATA3, CK5, CK7, SMA, GCDFP-15, bCL2).

[5] There is no explanation of what the numbers in parentheses in Table 1 are, e.g., when discussing Foshini et al. (2) it is written "CK7 (10/13), CK14 (3/13), etc.".

Author Response

Response to reviewer 2 comments

We thank the reviewers for their valuable comments that improved the content and readability of the manuscript. All edits have been addressed point by point as below. All changes in the manuscript have been made in red for easy identification.

Reviewer 2 comments

Comments and Suggestions for Authors

  • The manuscript by Elghobashy and co-workers focuses on the tall cell carcinoma of the breast with reversed polarity (TCCRP). The Authors describe a new case of this rare cancer and summarize the current knowledge about TCCRP. The article is interesting and well-written. I would suggest some changes as listed below.

We thank the reviewer for the positive feedback

  • [1] Please discuss the similarities/differences between TCCRP and papillary thyroid cancer in the context of the latest WHO histologic classification of thyroid tumors released in 2022. 

We have now discussed in detail the similarities and differences between TCCRP and papillary thyroid carcinoma (tall cell variant) including the morphological features, immunohistochemistry and molecular alterations. We have also cited the latest WHO book of the classification of thyroid tumours. The information has been inserted in the Discussion section (paragraph 2). We have also included an additional column with the characteristic features of papillary thyroid carcinoma to table 2 to highlight the similarities and differences with TCCRP and other malignant breast mimics.

  • [2] I wonder if the lump in this patient had a specific value on the BIRADS scale. If so, please provide this information in the manuscript.

The BIRADS score has been inserted in the Clinical Presentation section. The features were radiologically suspicious for malignancy (M4, U4).

  • [3] Gene names should be italicized (e.g., p. 4, line 123).

Genes name have now been capitalized throughout the manuscript.

  • [4] It seems to me that the Reader can expect a better explanation of the meaning/purpose of testing the various antigens mentioned in the manuscript in the context of breast/thyroid cancers (e.g. GATA3, CK5, CK7, SMA, GCDFP-15, bCL2).

We have now inserted a new section describing in detail the use of immunohistochemistry in establishing the diagnosis and differentiating from breast and thyroid mimics (Discussion section; paragraphs 1 &2). The immunohistochemical panel performed on the current case was based on the published literature of previously diagnosed cased including the WHO book for breast tumours. The immunohistochemistry is also highlighted in tables 1 and 2. A new reference has been inserted on the use of GATA3 immunohistochemistry in comparison with its rare expression in thyroid neoplasms.

  • [5] There is no explanation of what the numbers in parentheses in Table 1 are, e.g., when discussing Foshini et al. (2) it is written "CK7 (10/13), CK14 (3/13), etc.".

This has now been spelt out in the table. The numerator is the number of positive cases and denominator, number of total cases tested.

We hope we have addressed the reviewers’ comments satisfactorily and look forward to your decision.

Round 2

Reviewer 2 Report

The manuscript has been revised as suggested, so I recommend it for publication.